# Computational insights on the hydride and proton transfer mechanisms of L-proline dehydrogenase

**Ibrahim Yildiz** [ID] *

Chemistry Department and Applied Material Chemistry Center (AMCC), Khalifa University, Abu Dhabi, UAE

* ibrahim.yildiz@ku.ac.ae

## Abstract

L-Proline dehydrogenase (ProDH) is a flavin-dependent oxidoreductase, which catalyzes the oxidation of L-proline to (S)-1-pyrroline-5-carboxylate. Based on the experimental studies, a stepwise proton and hydride transfer mechanism is supported. According to this mechanism, the amino group of L-proline is deprotonated by a nearby Lys residue, which is followed by the hydride transfer process from C5 position of L-proline to N5 position of isoalloxazine ring of FAD. It was concluded that the hydride transfer step is rate limiting in the reductive half-reaction, however, in the overall reaction, the oxidation of FAD is the rate limiting step. In this study, we performed a computational mechanistic investigation based on ONIOM method to elucidate the mechanism of the reductive half-reaction corresponding to the oxidation of L-proline into iminoproline. Our calculations support the stepwise mechanism in which the deprotonation occurs initially as a fast step as result of a proton transfer from L-proline to the Lys residue. Subsequently, a hydride ion transfers from L-proline to FAD with a higher activation barrier. The enzyme-product complex showed a strong interaction between reduced FAD and iminoproline, which might help to explain why a step in the oxidative half-reaction is rate-limiting.

## 1. Introduction

Prokaryotes and eukaryotes oxidize proline into glutamate using two enzymes [1]. An FAD-based enzyme, proline dehydrogenase (ProDH), catalyzes the oxidation of proline into $\Delta^1$-pyrroline-5-carboxylate (P5C) while an NAD-based enzyme, P5C dehydrogenase (P5CDH), catalyzes the oxidation of glutamate semialdehyde into glutamate. In some gram-negative bacteria, the function of these two enzymes are linked into a single enzyme with two active sites known as proline utilization A (PutA) [1]. ProDH is associated with schizophrenia and related neurological disorders in humans, and its inhibition reduced the growth of breast tumor cells [2–5]. When the nutrients are not abundant, bacteria use proline as a source of carbon and nitrogen through PutA [1]. Therefore, selective inhibitors of enzymes related to proline metabolism and transport have potential therapeutic importance against some pathogens such as *helicobacter pylori*, *staphylococcus aureus*, and *mycobacterium tuberculosis* [6].

**Competing interests:** The authors have declared that no competing interests exist.

Steady-state kinetics data from investigating the ProDH activity of PutA from *Escherichia coli* towards L-proline showed a two-site ping-pong mechanism [7]. The first half of the reaction involves L-proline oxidation and formation of the reduced FAD cofactor, which is then oxidized by ubiquinone in the second half of the reaction. In order to elucidate the oxidation mechanism of L-proline, Serrano et al. performed steady-state kinetic, site-directed mutagenesis, and solvent and multiple kinetic isotope effect studies using ProDH from *Mycobacterium tuberculosis* [8]. The pH dependencies of k*cat* and k*cat*/K*pro* showed that a residue with a pKa of 6.8 should be deprotonated for the catalysis. Based on the structures of similar enzymes, this group was predicted to be either Lys110 or Tyr203 [9, 10]. Out of the two mutant enzymes, K110A did not show any catalytic activity while Y203F did not show any change in k*cat* [8]. This clearly pointed out Lys110 as the catalytic base. Hydride transfer mechanism has been shown to be a viable pathway for a variety of amine oxidases [11–13]. For amino acid oxidases, two oxidation mechanisms involving hydride transfer process, concerted or stepwise, have been commonly proposed [14–16]. According to the concerted mechanism, after the binding of amino acid to the active site, the deprotonation of α-amino group occurs simultaneously with the transfer of a hydride ion from α-C position of substrate to FAD. However, according to the stepwise mechanism, first deprotonation of α-amino group (1 in Fig 1) by an active site base occurs, then this is followed by the hydride transfer. ProDH is structurally different from the other amino acid oxidases. First of all, the substrate is a secondary amine as opposed to a primary amine, and secondly the oxidation process does not involve transfer of a hydride ion from α-C rather from C5 position (2 in Fig 1). Based on this, a reduced FAD (FADH⁻ in Fig 1) and iminoprolinie intermediate (3 in Step 2 in Fig 1) are produced. The primary kinetic isotope effect studies in water and deuterium oxide showed that ProDH follows a stepwise mechanism, and proton transfer is faster than hydride transfer step [8].

Up to now, there is no report on the crystal structure of the ProdH complexed with either L-proline or the intermediate iminoproline. Zhang et al. resolved the crystal structures of *Escherichia coli* proline utilization A (PutA) in which ProDH domains were complexed with competitive inhibitors such as acetate, L-lactate, and L-tetrahydro-2-furoic acid (L-THFA) [9]. These structures provided clues about the roles of active site residues during the binding of substrate and catalysis. Among these inhibitors, L-THFA is structurally very similar to L-proline, and the only difference is O atom instead of α-N (Fig 2). In the complex, L-THFA is sandwiched between FAD and active site residues. O atoms of carboxylate group of L-THFA have close distances to two N atoms of the guanido group of Arg555. In addition, another Arg residue, Arg556, is close to one of the O atoms of carboxylate group suggesting H-bonding

**Fig 1. The oxidation of L-proline (1) into iminoproline (3) by ProDH through stepwise proton and hydride transfer processes.** An active site species acts as proton acceptor while FAD acts as hydride acceptor.

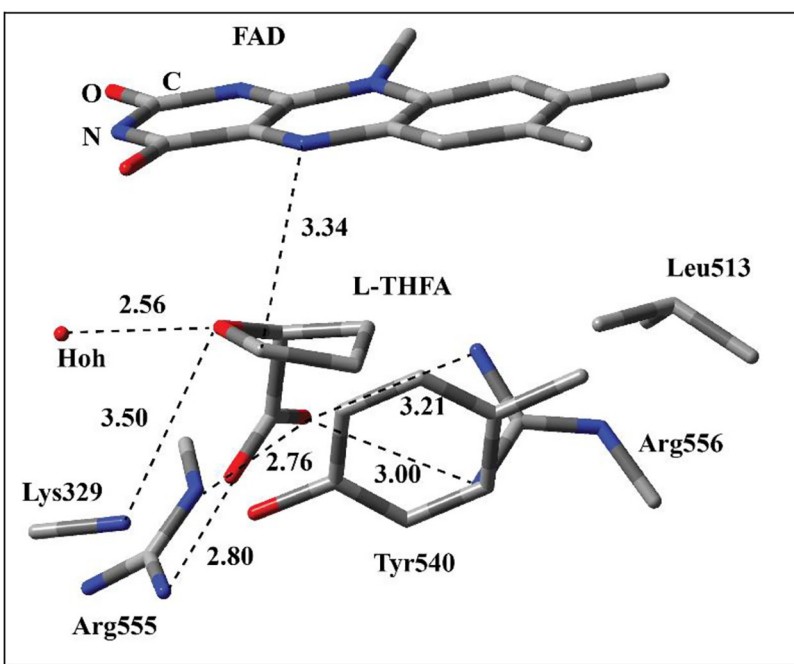

**Fig 2. The active site of ProDH domain of *Escherichia coli* proline utilization A (PutA) enzyme complexed with L-THFA inhibitor including FAD and the surrounding residues.** The chemical interactions between L-THFA and surrounding residues and the distance between C5 atom of L-THFA and N5 position of FAD are shown with the dotted lines. The distances are in Å.

interactions. These interactions suggest that two positively charged Arg residues play an important role in the binding of substrate due to their interaction with the carboxylate group. Furthermore, Lys329 and a water molecules have close distances to the O atom of the ether group of L-THFA suggesting H-bonding interactions. From this observation, it could be deduced that Lys329 may act as the catalytic base to deprotonate the α-amino group of L-proline in the enzyme-substrate complex. In addition, the aliphatic ring of L-THFA is surrounded by Tyr540 and Leu513 suggesting hydrophobic interactions. All these interactions seem important for the binding and orientation of substrate, L-proline, in the active site, and they might play important roles during the hydride and proton transfer steps. The distance between C5 of L-THFA and N5 atom of isoalloxazine ring is 3.34 A°, which implies that isoalloxazine ring and L-proline may have proper distance and orientation for the hydride transfer process.

Computational methods offer invaluable mechanistic insights for the enzymes lacking resolved crystal structures with either native substrates, intermediates, or products. ONIOM method including QM-MM (quantum mechanics-molecular mechanics) calculations has started to be an attractive tool to study the details of enzyme mechanisms [17–22]. This method was successfully used to analyze a number of amine oxidases and dehydrogenases [23]. QM component could elucidate important information related to chemical transformations, and MM component could provide a more realistic protein environment around the active site.

In this study, the hydride and proton transfer steps in the oxidation of L-proline to imino-proline by ProDH (Fig 1) was investigated using ONIOM methods. Model systems were built using the crystal structure of the enzyme-L-THFA complex (Fig 2). Through model enzyme-substrate complex, transition state, and enzyme-product complexes, the roles of active site residues before, during, and after the hydride and proton transfer processes were highlighted.

## 2. Computational details and methodology

In the calculations, we used a two-layer ONIOM system [18] consisting of a QM region which was treated with three common DFT functionals—CAM-B3LYP [24], M06-2X [25], and ωB97XD [26],—and an MM region which was treated with AMBER force field [27]. Gaussian 09 package was used to run the calculations [28] CAM-B3LYP functional was chosen due to its hybrid quality of B3LYP functional with long-range correction [17, 29]; M06-2X functional was chosen due to its better performance in the main-group elements in comparison with B3LYP [30]; ωB97XD functional was chosen since it provides empirical dispersion and long-range corrections. The missing MM charges for all atoms in FAD were calculated as RESP charges using HF method with 6-31G(d) basis set. The missing MM parameters for FAD were derived using antechamber option in AMBER 16 Tools [31, 32]. Amber 94 MM charges were the defaults charges for all atoms in the other residues and water molecules. Mechanical embedding of MM region into QM region was included in the calculations.

The geometries of the reactant complex (RC,) product complex (PC), and transition state (TS) were optimized using 6-31G basis set for three DFT functionals and with 6-31G(d,p) basis set for M06-2X functional. Single point energy calculations were performed using 6–311 +g(2d,2p) basis set for the species optimized with 6-31G(d,p) basis set with M06-2X functional. The optimized TS structures are validated with frequency calculations requiring one negative eigenvalue, and RC and PC were validated without any negative eigenvalues. Frequency calculations were done at 25°C and 1 atm. TS structures were validated with intrinsic reaction coordinate (IRC) calculations [33]. TS structures were located through potential energy surface (PES) scans by scanning the bond coordinates, and the maximum energy points in the scans were subjected to TS optimization using Berny algorithm [34].

The model enzyme-substrate complex included FAD, L-proline, and the residues located around L-proline in a radius of 10 Å. The model enzyme-L-proline complex was obtained from the crystal structure of the enzyme bound with L-THFA (PDB accession code: 1tiw) [9] using VMD program [35]. L-THFA was converted to L-proline by changing the ether O atom into N atom. This model consists of 1279 atoms, 81 residues, L-proline, FAD, and 25 water molecules. N-terminal and C-terminal residues on the peripheries were connected with acetyl and N-methyl groups to maintain the electrostatic environment of the original active site as well as to prevent extra charged residues around the active site. The protonation states of the residues with ionizable side groups were decided using PropKa program [36]. All H atoms were added with GaussView 5 program. The total charge of model systems was -3. In the model systems, QM region included FAD, L-proline, Arg555, Arg556, and Lys329. All the other residues were included in the MM region. The QM region included 77 atoms with a total charge of +2. FAD and active site residues were partitioned into QM and MM regions.

The initial structure was optimized to produce a model enzyme-L-proline complex. The TS structure for the proton transfer step between L-proline and Lys residue, Lys329, was located through a PES scan performed on the optimized model enzyme-product complex using the coordinate shown with the dashed line labeled as "a" in Fig 3. The TS structure for the hydride transfer step between L-proline and FAD was located through a PES scan performed on the optimized model enzyme-product complex using the coordinate shown with the dashed line labeled as "b" in Fig 3. The distances were decreased over a number of steps. The geometry of the highest energy points in the PES scans were optimized to obtain TS structures. The RC and PC of the hydride transfer process and proton transfer processes were optimized using proper geometries in the PES scan before and after highest energy point. RC and PC obtained from the PES scan were confirmed with the IRC calculation. The optimized geometries of the proton transfer step are labeled as RC1, TS1, and PC1 in the following section. The optimized

**Fig 3. The coordinates used for the PES scan to locate the TS structures for the hydride and proton transfer processes for enzyme-L-proline complex.**

geometries of the hydride transfer step are labeled as RC2 (same as PC1), TS2, and PC2 in the following section.

## 3. Results and discussions

The PES scan for the hydride transfer process involving N5 position of the isoalloxazine ring of FAD and H atom at the C5 atom of L-proline (a in Fig 3) indicated that the proton transfer process occurs before the hydride transfer process. As the hydride ion moves along this coordinate, the proton at the $\alpha$-amino group of L-proline transfers itself to Lys329 before hydride transfer step reaches its transition state. In order to obtain the structures of the reactant (RC1), TS (TS1) and product (PC1) of the proton transfer step, we performed another PES scan (b in Fig 3) involving this process. In the coming sections, we will discuss first proton transfer step, then the hydride transfer step.

### 3.1. ONIOM model systems for proton transfer mechanism

The optimized geometry of the model RC1 corresponding to reactive enzyme-substrate complex shows that L-proline binds to the active site of ProDH as its $\alpha$-amino group protonated

(Fig 4). This is in agreement with the previous findings that the amine-based substrates prefer to bind to amine oxidases and dehydrogenases as their amino group protonated [37]. The iso-alloxazine ring of FAD has a planar conformation. In this structure, the amino group in the side chain of Lys329 is neutral as it was proposed to be the catalytic base deprotonating the amino group of L-proline [8]. Indeed, there is a close H-bonding interaction (1.83 Å) between the N atom of Lys329 and H atom at the amino group of L-proline suggesting that it might act as the base. The other H atom of the amino group of L-proline has H-bonding interaction with the carbonyl O atom of the isoalloxazine ring of FAD. The guanido groups of two arginine residues, Arg555 and Arg556, have H-bonding interactions with the carboxylate group of L-proline. This shows that the positively charged Arg residues form ion pairs with the negatively charged carboxylate group. In addition to these interactions, L-proline is surrounded with FAD and three other residues (Tyr540, Tyr552, and Leu513) which are located in the MM region. This suggests that these three residues have hydrophobic interactions with L-proline, and MM method can model the chemical interactions properly. There is also a layer of other residues in the MM region interacting with the subtrate, FAD, and the residues in QM region.

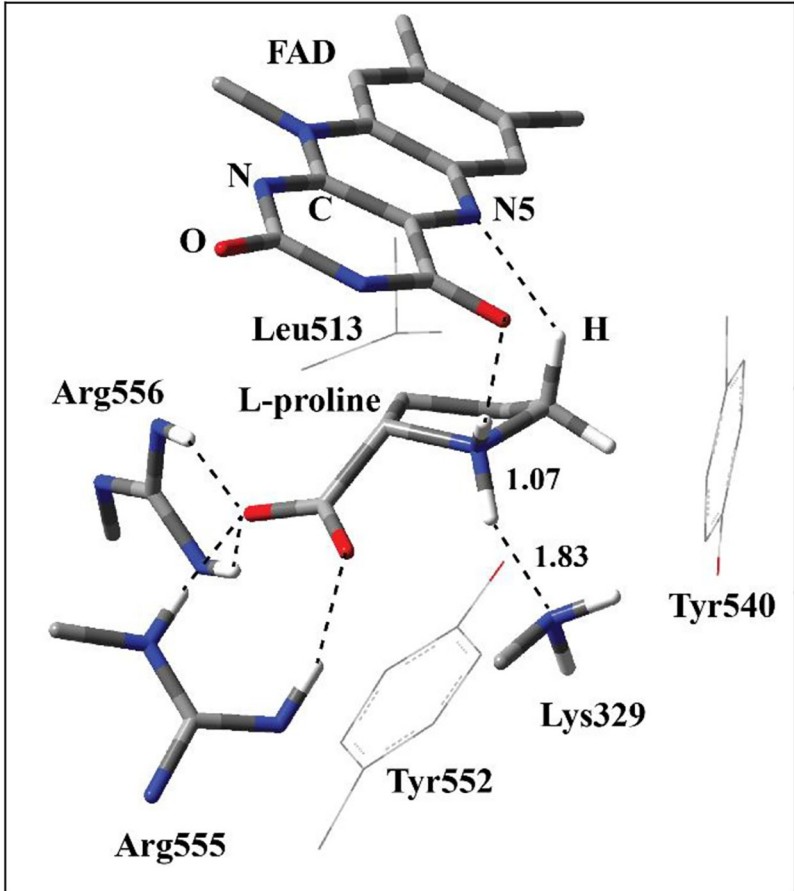

**Fig 4. The structure of optimized RC1 including FAD, L-proline, Arg555, Arg556, and Lys329 in QM region and Tyr540, Tyr552, and Leu513 in MM region obtained with ONIOM(M06-2X /6-31G(d,p):Amber).** QM region is shown with tube models while three MM region residues are shown with wireframe model. All the H atoms are excluded in the figure for clarity except the ones shown with ivory color. The H bonding interactions and the distance between N5 position of the isoalloxazine ring of FAD and H atom at C5 position of L-proline are shown with the dashed lines. The distances are in Å.

For clarity reasons, they are not shown in Fig 4. These interactions can be analyzed using text files including the Cartesian coordinates of RC, TS, and PC species deposited as S1 Data.

The optimized TS1 structure for the proton transfer step (A in Fig 5) shows that the proton is in flight from amino group of L-proline to Lys329. The distance between the proton and Lys329 decreased as compared to RC1. Similar to RC1, two arginine residues have interactions with the carboxylate group of L-proline. L-proline has H-bonding interaction with one of the carbonyl O atom of isoalloxazine ring. The animation of imaginary frequency for the TS1 (S1 Movie) shows the transfers of proton from L-proline to FAD. In the optimized structure of PC1 (B in Fig 6), the proton transferred to Lys329 completely. In addition to H-bonding between Lys329 and L-proline, similar H-bonding interactions exist as in the case of RC1 and TS1.

The absolute energy difference between TS1 and RC1 corresponding to the activation energy for the proton transfer process from L-proline to Lys329 in terms of ONIOM energy was calculated 4.32 kcal/mol with M06-2X functional with 6-31G(d,p) basis set in QM region (G-E$_{ar}$ in Table 1). A slightly higher activation energy barrier (5.93 kcal/mol) was calculated with 6–311++G(2d,2p) basis set in QM region. With the same basis set, CAM-B3LYP and ωB97XD estimated 1–2 kcal/mol more energies than M06-2X. All three functionals estimated smaller activation energies (0.39–1.55 kcal/mol) for the reverse process. Both forward and reverse activation energy values are small, and they indicate low barriers for the proton transfer process. The entropic and enthalpic contribution lowers the activation energy barrier. Activation barriers in terms of absolute energy, zero-point corrected absolute energy, enthalpy, and Gibbs free energy are provided in the S6-S9 Tables in S1 File.

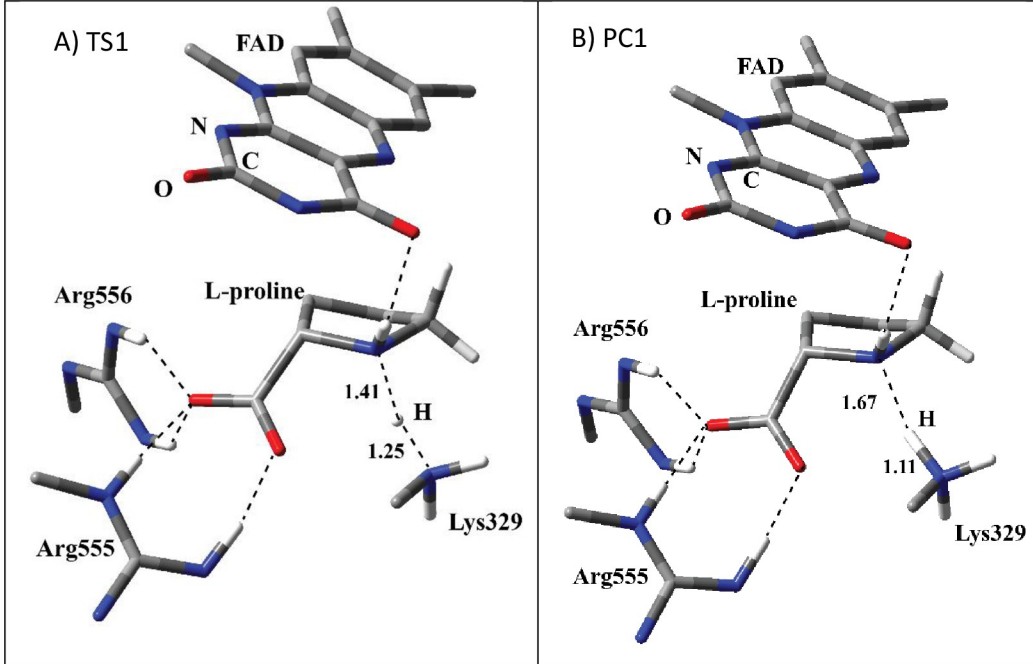

**Fig 5. The structure of optimized TS1 (A) and PC1 (B) including FAD, L-proline, Arg555, Arg556, and Lys329 in QM region obtained with ONIOM(M06-2X /6-31G(d,p):Amber).** QM region is shown with tube models. All the H atoms are excluded in the figure for clarity except the ones shown with ivory color. The H bonding interactions and bond breaking/forming processes are shown with the dashed lines. The distances are in Å.

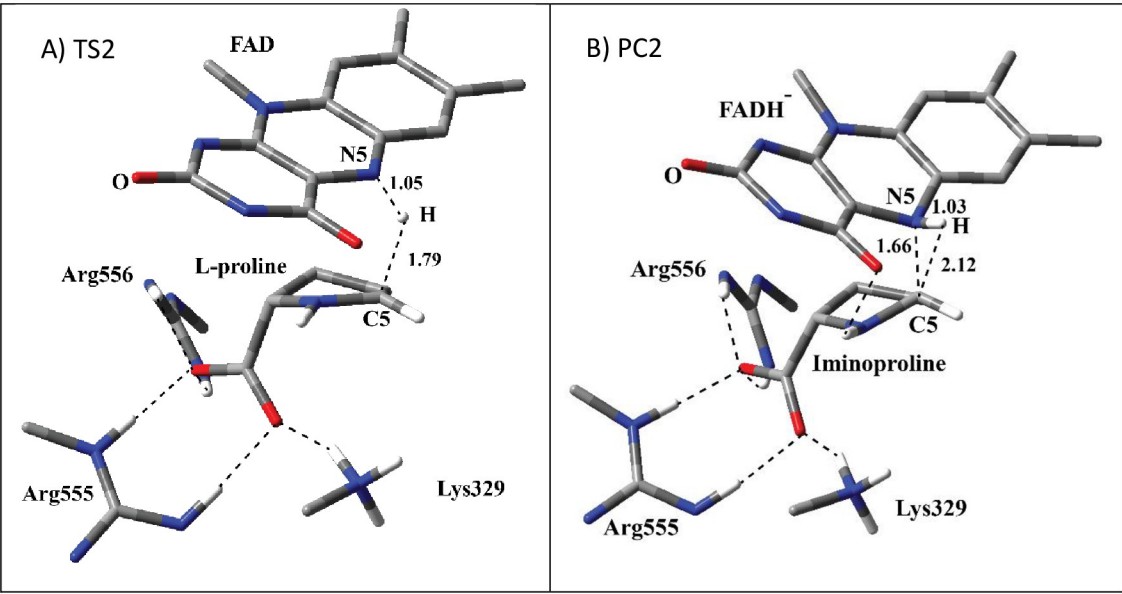

**Fig 6. The structure of optimized TS2 (A) and PC2 (B) including FAD, L- and imino-proline, Arg555, Arg556, and Lys329 in QM region obtained with ONIOM(M06-2X /6-31G(d,p):Amber).** QM region is shown with tube models. All the H atoms are excluded in the figure for clarity except the ones shown with ivory color. The H bonding interactions and bond breaking/forming processes are shown with the dashed lines. The distances are in Å.

### 3.2. ONIOM model systems for hydride transfer mechanism

PES scans using the bond coordinate between N5 atom at isoalloxazine ring and H atom at C5 position of L-proline before (RC1 in Fig 4) or after (PC1 in Fig 5) proton transfer process produced the same TS2 structure for the hydride transfer process (A in Fig 6). As discussed previously, along the PES scan coordinate involving RC1 proton transfer occurs before hydride transfer process. The optimized TS2 structure (A in Fig 6) shows that the hydride ion moves

**Table 1. Energy profile for the hydride and proton transfer processes for ONIOM calculations using CAM-B3LYP, M062X, ωB97XD functionals in the QM region with 6-31G, 6-31G(d,p), and 6–311+G(2d,2p) basis sets.** (G-E$_{af}$: activation energy for the forward reaction in kcal/mol in terms of ONIOM Gibbs free energy, G-E$_{ar}$: activation energy for the reverse reaction in kcal/mol in terms of ONIOM Gibbs free energy, E-E$_{af}$: activation energy for the forward reaction in kcal/mol in terms of ONIOM absolute energy, G-E$_{ar}$: activation energy for the reverse reaction in kcal/mol in terms of ONIOM absolute energy, ND: Not determined). The activation energy values with 6–311+G(2d,2p) basis set were obtained through single point energy calculations on the structures optimized with 6-31G(d,p) basis set with M062X.

| | Functional | Basis Set | Ea$_f$(kcal/mol) | | Ea$_r$(kcal/mol) | | *i* |
|---|---|---|---|---|---|---|---|
| | | | G | E | G | E | |
| **Proton Transfer Step** | M062X | 6-31G(d,p) | 1.46 | 4.32 | -0.47 | 0.39 | -662.49 |
| | | 6–311+G(2d,2p) | ND | 5.93 | ND | 0.94 | ND |
| | CAM-B3LYP | 6–311+G(2d,2p) | ND | 6.99 | ND | 1.44 | ND |
| | ωB97XD | 6–311+G(2d,2p) | ND | 7.27 | ND | 1.55 | ND |
| **Hydride Transfer Step** | M062X | 6-31G | 20.79 | 25.49 | 19.70 | 25.46 | -535.11 |
| | | 6-31G(d,p) | 25.28 | 30.18 | 17.67 | 23.32 | -373.35 |
| | | 6–311+G(2d,2p) | ND | 27.37 | ND | 21.37 | ND |
| | CAM-B3LYP | 6-31G | 23.86 | 27.79 | 17.49 | 22.64 | -531.22 |
| | | 6–311+G(2d,2p) | ND | 28.75 | ND | 19.64 | ND |
| | ωB97XD | 6-31G | 20.23 | 24.97 | 18.18 | 23.10 | -455.76 |
| | | 6–311+G(2d,2p) | ND | 26.82 | ND | 19.93 | ND |

from C5 atom of L-proline to N5 position of isoalloxazine ring of FAD. The animation of imaginary frequency for TS2 S2 Movie shows this process. The N atom of α-amino group and C5 atoms in L-proline start to assume sp$^2$ character as L-proline starts to convert to iminoproline. In TS2, the hydride ion is much closer to N5 of isoalloxazine ring as compared to C5 of L-proline (1.05 Å vs 1.79 Å) suggesting that the TS structure resembles more to product. In TS2, Lys329 does not have any more H-bonding interaction with of the N atom of α-amino group at L-proline. It has H-bonding interaction with the carboxylate group together with Arg555 and Arg556 residues.

The optimized structure of PC2 (B in Fig 6) shows that the hydride transfer process is complete, and L-proline is converted into iminoproline together with the formation of reduced FAD. Similar H-bonding interactions exist as in the case of TS2. The isoalloxazine ring assumed bent conformation. One quite striking observation is that the geometry around iminium bond is not planar. Indeed, imino N and C atoms in iminoproline seems to have more tetrahedral character rather than planar. The distance between N5 atom of isoalloxazine and C5 position of iminoproline is 1.66 Å suggesting that there is a strong dipole-dipole interaction between N5 and C5 atoms. The distance does not suggest a full covalent bond, however, the interaction seems to be closer to a strong coordination interaction. N5 position of reduced FAD acts as the nucleophile and C5 position of iminoproline act as the electrophile to form reduced FAD-iminoproline adduct. In the literature, it was shown that N5 position of reduced FAD is capable of nucleophilic attack to form covalent bonds with a variety of substrates [38, 39]. For the enzyme turnover, the reduced FAD needs to be oxidized in the oxidative half-reaction, and the strong interaction between iminoproline and reduced FAD might have caused an increase for the product release. In order to test whether this strong interaction between reduced FAD and iminoproline is driven by reduced FAD, the H atom at N5 of reduced FAD was removed to simulate the oxidation of reduced FAD. The resulting model was optimized using 6-31G(d,p) basis set with M06-2X functional. The optimized model (S1 Fig in S1 File) revealed that the iminium bond becomes planar in iminoproline, and there is no more a strong interaction between N5 atom of isoalloxazine ring of oxidized FAD and C5 position of iminoproline. It is also visible that imimoproline and FAD separated after the oxidation process. It is also quite possible that some conformational changes in the active site might disrupt this interaction between iminoproline and reduced FAD, and as a result iminoproline might leave the active site. Future MD simulations can shed more light in this regard.

The Gibbs free energy difference between TS2 and RC2 (PC1) corresponding to the activation energy for the hydride transfer process (G-E$_{ar}$ in Table 1) in terms of ONIOM energy was calculated 20.79 kcal/mol with M06-2X functional with 6-31G basis set. This value is estimated 23.86 kcal/mol with CAM-B3LYP functional, and 20.23 kcal/mol with ωB97XD functional. A larger basis set (6-31G(d,p)) with M06-2X functional estimated almost 5 kcal/mol more activation barrier. The values of the activation barriers for hydride transfer process in terms of absolute energy are more than the values in terms of Gibbs free energy indicating that enthalpic and entropic terms lower the activation barrier (S6-S9 Tables in S1 File). Activation barrier in terms of absolute energy based on single point energy calculation with 6–311+G(2d,2p) basis set with M06-2X functional was approximately 2 kcal/mol more than the one obtained with 6-31G. Single point energy calculations with 6–311+G(2d,2p) basis set with CAM-B3LYP and ωB97XD functionals also showed higher activation barriers as compared to 6-31G basis set. Previously higher activation barriers were observed with larger basis sets for ONIOM calculations with other FAD-based amine oxidases [23, 40]. Based on the reported k*red* value of FAD for ProDH from *Escherichia coli* (27.5 s$^{-1}$) [41], an activation barrier of 16.5 kcal/mol can be calculated for the hydride transfer step using Arrhenius equation. Therefore, the activation barriers in terms of Gibbs free energy with smaller basis set, 6-31G, are better estimates than

the values obtained with the larger basis sets. In addition, computational studies seem to over-estimate the activation barriers for the hydride transfer process for a number FAD-based oxidases [42–44].

## 4. Conclusion

In this study, the reductive half-reaction corresponding to hydride transfer step and proton transfer step for the oxidation of L-proline to iminoproline by ProDH was investigated using ONIOM method through hybrid QM-MM calculations. The model enzyme-L-proline complex was obtained from the crystal structure of the enzyme complexed with a substrate analog. Through this model system, we formulated enzyme-cofactor-substrate complex, TS structures for the hydride and proton transfer steps, and enzyme-cofactor-product complex. It was shown that proton transfer step occurs with a very small barrier from L-proline to a nearby Lys residue. In the following step, a hydride ion moves from C5 position of L-proline to N5 position of FAD forming iminoproline and reduced FAD. It was found that H-bonding interactions as well as hydrophobic interactions, before, during, and after the proton and hydride steps are important for binding and catalysis. We also found that iminoproline exhibits a strong interaction with reduced FAD. The mechanistic interpretation of ProdH may lead to development of novel inhibitors against a number of diseases. In future, oxidative half-reaction can be studied with computational methods to obtain further insights into mechanism of ProdH.

## Supporting information

**S1 File.**
(DOCX)

**S1 Data.**
(ZIP)

**S1 Movie.**
(MP4)

**S2 Movie.**
(MP4)

**S1 Graphical abstract.**
(DOCX)

## Acknowledgments

The authors acknowledge the contribution of High-Performance Computing Facility and Applied Material Chemistry Center (AMCC) at Khalifa University.

## Author Contributions

**Conceptualization:** Ibrahim Yildiz.

**Formal analysis:** Ibrahim Yildiz.

**Investigation:** Ibrahim Yildiz.

**Methodology:** Ibrahim Yildiz.

**Visualization:** Ibrahim Yildiz.

**Writing – original draft:** Ibrahim Yildiz.

**Writing – review & editing:** Ibrahim Yildiz.

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
