## [Decision Letter · Decision Letter 0]

28 Jul 2023

PONE-D-23-19955Computational Insights on the Hydride and Proton Transfer Mechanisms of L‑Proline DehydrogenasePLOS ONE

Dear Dr. yildiz,

Thank you for submitting your manuscript to PLOS ONE. After careful consideration, we feel that it has merit but does not fully meet PLOS ONE’s publication criteria as it currently stands. Therefore, we invite you to submit a revised version of the manuscript that addresses the points raised during the review process.

We look forward to receiving your revised manuscript.

Kind regards,

Kshatresh Dutta Dubey

Academic Editor

PLOS ONE

Journal Requirements:

Reviewers' comments:

Reviewer's Responses to Questions

**Comments to the Author**

1. Is the manuscript technically sound, and do the data support the conclusions?

Reviewer #1: Partly

Reviewer #2: Yes

2. Has the statistical analysis been performed appropriately and rigorously? 

Reviewer #1: N/A

Reviewer #2: Yes

3. Have the authors made all data underlying the findings in their manuscript fully available?

Reviewer #1: Yes

Reviewer #2: Yes

4. Is the manuscript presented in an intelligible fashion and written in standard English?

Reviewer #1: Yes

Reviewer #2: Yes

5. Review Comments to the Author

Reviewer #1: Major comments:

(1) The author choose several DFT functionals and basis sets in the calculations. However, the author comes to a conclusion that " the activation barriers in terms of Gibbs free energy with smaller basis set, 6-31G, are better estimates than the values obtained with the larger basis sets. " . This conclution is surprising, because the 6-31G basis set do not gives a proper description to hydrogen, thus larger error would be expected when concerning a reaction with hydrogen element directly involved in TS geometry. More careful calculations should be conducted before the author address this conclusion.

(2) The author should give a reasonable explanation about why the bigger basis set singly-point calculations were only performed for M06-2X? Based on so many published benchmark studies, the energies obtained from a small basis set like 6-31G were expected to give a big error, the optimization should be performed at least on 6-31G** level, and Single-Point calculations for other reported functionals are required.

(3) For the description of "The activation barriers for hydride transfer process in terms of absolute energy are 3-10 kcal/mol more than in terms of Gibbs free energy indicating that enthalpic and entropic terms lower the activation barrier. ". The affect of the enthalpic and entropic correction can be directly read from the log file of a frequency calculation, the author should provide exact numbers at here.

Minor comments:

(1) Too many background information about the PRODH was mentioned in the Abstract section of the manuscript.

Reviewer #2: The authors provide a well done computational study of the mechanism of proline oxidation by the flavin-dependent enzyme proline dehydrogenase. The findings will be of significant interest to the field as the results provide unique insights into mechanistic possibilities for ProDH. The manuscript is well written and the results are explained in a succinct manner. Some recommendations are suggested below.

Comments

1. Abstract- Line 1: Replace “member of amino acid oxidase” with is “flavin-dependent oxidoreductase”

2. Abstract- line 14: Replace “supported” with “support”

3. Page 2, line 3 from the bottom: Replace “L-proline oxidation is followed by the reduction of FAD through ubiquinone.” With “The first half of the reaction involves L-proline oxidation and formation of the reduced FAD cofactor, which is then oxidized by ubiquinone in the second half of the reaction.”

4. Page 2, line 4 from the bottom: Replace “involving the activity of ProDH” with “from investigating the ProDH activity”

5. Page 7, line 13 “L-proline binds to the active site”

6. Figure 4 shows the modeling of proline in active site with the amino group protonated. Did the authors try modeling proline without the amino group protonated? Or can only the protonated form of proline be modeled active site?

7. Provide labels for the Figures. For example, in Figure 5 label panel A as “TS1” and panel B as “PC1”. Figure 6, panel A “TS2” and panel B “PC2”.

8. The distance between the N-atom of Lys and amino group of proline is calculated to decrease upon going from RC1 and TS1. Is there also a change in distance between the carbonyl O atom of the FAD and the proline amino group between RC1 and TS1?

9. The authors provide an analysis of the different energy barriers for the proton and hydride transfer steps. Can the binding energies of the substrate/product molecules in the RC1, TS1 and PC1 also be estimated from the modeling?

10. Page 9, line 4 from the bottom: “compared to the C5 of L-proline”

11. The illustration in Figure 6, panel A seems to show that the hydride transfer from proline to the FAD N5 does not follow a linear path. What is the direct distance between the C5 of proline and the N5 of FAD? Is the purpose of this figure to show the projection of the hydride from the N5 of FAD or is it supposed to show the hydride being shared between the C5 and N5 atoms?

12. Page 11, Line 6: The authors comment “Based on this observation, it can be

inferred that the release of iminoproline from the active site may occur after the oxidation of

reduced FAD.” If this happened, it would imply a ternary mechanism, with P5C and ubiquinone bound to the enzyme simultaneously. This conclusion would contrast with experimental data showing a ping-pong mechanism. The authors perhaps should modify this statement or remove this conclusion until they conduct computational studies of the oxidative half-reaction.

13. In regards to the PC2 complex shown in Figure 6A, can the authors provide further insights into the possibility of the reverse step of hydride transfer from reduced FAD to iminoproline, thereby generating proline and oxidized FAD? What activation barrier would there be for this? Besides implying covalent bond formation with the N5, could the PC2 complex also imply the possibility of the reaction being reversible in the absence of ubiquinone?

14. Page 10, lines 10-12. Reference 38 is used to support the possibility of N5 nucleophilic attack on substrates. In addition to this review, a specific example of N5 attack on a covalent inhibitor for a ProDH enzyme is provided by the following study:

Srivastava D, Zhu W, Johnson WH Jr, Whitman CP, Becker DF, Tanner JJ. The structure of the proline utilization a proline dehydrogenase domain inactivated by N-propargylglycine provides insight into conformational changes induced by substrate binding and flavin reduction. Biochemistry. 2010 Jan 26;49(3):560-9. doi: 10.1021/bi901717s. PMID: 19994913; PMCID: PMC3727237.

This reference would be useful to include as well.

6. PLOS authors have the option to publish the peer review history of their article (what does this mean?). If published, this will include your full peer review and any attached files.

Reviewer #1: No

Reviewer #2: No

---

## [Author Response · Author response to Decision Letter 0]

8 Aug 2023

A separate document was uploaded addressing the reviewers comments.

---

## [Editor Report · Decision Letter 1]

18 Aug 2023

Computational Insights on the Hydride and Proton Transfer Mechanisms of L‑Proline Dehydrogenase

PONE-D-23-19955R1

Dear Dr. yildiz,

We’re pleased to inform you that your manuscript has been judged scientifically suitable for publication and will be formally accepted for publication once it meets all outstanding technical requirements.

Kind regards,

Kshatresh Dutta Dubey

Academic Editor

PLOS ONE
---

## [Editor Report · Acceptance letter]

31 Aug 2023

PONE-D-23-19955R1 

Computational Insights on the Hydride and Proton Transfer Mechanisms of L‑Proline Dehydrogenase 

Dear Dr. Yildiz:

I'm pleased to inform you that your manuscript has been deemed suitable for publication in PLOS ONE. Congratulations! Your manuscript is now with our production department. 

Kind regards, 

on behalf of

Dr. Kshatresh Dutta Dubey 

Academic Editor

PLOS ONE